# Microsatellite Instability and Myometrial Infiltration in Low-Grade Endometrial Cancer: A Focus on MMR Heterodimer Dysfunction by a Retrospective Multicentric Italian Study

**DOI:** 10.3390/jpm15090417

**Published:** 2025-09-02

**Authors:** Carlo Ronsini, Stefano Restaino, Mariano Catello Di Donna, Giuseppe Cucinella, Maria Cristina Solazzo, Pasquale De Franciscis, Giuseppe Vizzielli, Manuela Ludovisi, Vito Chiantera

**Affiliations:** 1Unit of Gynecologic Oncology, National Cancer Institute, IRCCS, Fondazione “G. Pascale”, 80131 Naples, Italy; mariano.didonna@istitutotumori.na.it (M.C.D.D.); giuseppe.cucinella@istitutotumori.na.it (G.C.); mariacristinasolazzo@gmail.com (M.C.S.); vito.chiantera@istitutotumori.na.it (V.C.); 2Unit of Obstetrics and Gynecology, “Santa Maria della Misericordia” University Hospital, Azienda Sanitaria Universitaria Friuli Centrale, 33100 Udine, Italy; restaino.stefano@gmail.com (S.R.); giuseppevizzielli@yahoo.it (G.V.); 3Unit of Gynaecology and Obstetrics, Department of Woman, Child and General and Specialized Surgery, University of Campania “Luigi Vanvitelli”, 80138 Naples, Italy; pasquale.defranciscis@unicampania.it; 4Department of Life, Health and Environmental Sciences, University of L’Aquila, 67100 L’Aquila, Italy

**Keywords:** endometrial carcinoma, microsatellite instability (MSI), myometrial infiltration, low-grade endometrial cancer, fertility-sparing treatment

## Abstract

**Background:** Recent studies highlight the role of microsatellite instability (MSI) in tumor progression. This study examines the link between MSI, type of loss of function, and disease progression in low-grade endometrial carcinoma clinically confined to the uterus, focusing on myometrial infiltration. **Materials and Methods:** This retrospective case-control study analyzed data from 144 women treated for clinical stage I low-grade endometrial carcinoma at two university hospitals. Patients were divided into two groups based on microsatellite status: 118 with microsatellite stability (MSS) and 26 with MSI. Immunohistochemical profiling assessed MMR proteins (MLH1, PMS2, MSH2, MSH6). The primary outcome was the presence of myometrial infiltration, and the secondary outcome was the deepness of infiltration. Data were statistically analyzed using Fisher’s exact, Chi-square, and Wilcoxon tests, with logistic regression applied to evaluate the impact of MSI on these outcomes. **Results:** Myometrial infiltration was present in 96% of MSS and 98% of MSI cases (*p* = 0.5). However, deep infiltration (≥50%) was more frequent in patients with MSI (38% vs. 19%, *p* = 0.042). Stratification by heterodimer loss revealed that loss of MLH1/PMS2 was associated with a higher rate of deep infiltration (47%), while loss of MSH2/MSH6 correlated with lower infiltration risk (14%). In multivariate analysis, MSH2/MSH6 loss remained negatively associated with infiltration (OR 0.88; 95% CI 0.80–0.98; *p* = 0.020), independent of grade and LVSI. **Conclusions:** In low-grade endometrial carcinomas clinically confined to the uterus, MSI does not increase the overall prevalence of myometrial infiltration but is associated with deeper invasion, especially in cases with MLH1/PMS2 loss. MSI profiling could aid in risk stratification and therapeutic planning, particularly in candidates for fertility-sparing treatment.

## 1. Introduction

Endometrial carcinoma (EC) is the leading gynecological oncological disease by incidence in Western countries [1]. It is also the only disease whose incidence and mortality are increasing [2]. Recently, the scientific community’s efforts have focused on better elucidating the molecular aspects of this pathology to discriminate clinical and anatomopathological presentations with different prognostic impacts [3,4]. Among these, tumors with microsatellite instability (MSI) have garnered attention due to their distinct biology, immune environment, and potential responsiveness to immune checkpoint inhibitors [5]. Within the same histotype, microsatellite instability could influence a tumor’s susceptibility to progression, highlighting its role in prognosis and therapeutic stratification. We specifically focused on MMR deficiency as a molecular driver of microsatellite instability. Other molecular alterations such as p53 mutations or POLE mutations were not included, as they represent distinct categories in the TCGA/ProMisE classification and their inclusion would have introduced confounding effects. Understanding these nuances is critical for improving diagnostic precision and tailoring treatment strategies. For example, low-risk conditions in the absence of myometrial infiltration are candidates for fertility-sparing treatment (FST). However, the absence of infiltration is based on imaging and is only presumptive. The question of our study is to assess whether microsatellite instability (MSI) may increase the risk of myometrial infiltration in pathologies considered low risk (endometrioid histotype, grading 1, 2).

### Objective

This study aimed to investigate the association between microsatellite instability (MSI) and histologically confirmed myometrial infiltration in patients with low-grade endometrioid endometrial carcinoma (EC) and no radiological evidence of extra-uterine spread. Specifically, we sought to determine whether the presence of MSI correlates with a higher risk of myometrial invasion in tumors typically classified as low risk. Furthermore, a sub-analysis was performed to explore the relationship between the molecular mechanisms underlying MSI, namely the specific gene alterations causing loss of mismatch repair function, and the extent of myometrial infiltration. This approach aimed to clarify whether different patterns of MMR deficiency have distinct impacts on tumor invasiveness, thus contributing to a more refined risk stratification even in clinically indolent cases.

## 2. Material and Methods

### 2.1. Ethical or Institutional Review Board Approval

This study was conducted in two university clinics where all patients treated must sign a dedicated consent for anonymous data processing. The research methods were established a priori and authorized through evaluation by the Ethics Committee Università degli Studi della Campania “Luigi Vanvitelli”-Azienda ospedaliera Universitaria “Luigi Vanvitelli”-AORN “Ospedale dei Colli”, Campania 2 (IRB 30661/2022 of 31 March 2022).

### 2.2. Study Design

In January 2025, we conducted a retrospective observational multicenter cohort study with secondary data from the clinical database, selecting a total of 144 patients with EC who underwent primary staging surgery at the gynecologic oncology unit of the University of Campania Luigi Vanvitelli, Naples, Italy and the unit of obstetrics and gynecology, “Santa Maria della Misericordia” University Hospital, Udine, Italy. The research methods were established a priori. This study is a retrospective case-control analysis of women affected by low-grade endometrial endometrioid cancer, with clinical absence of extrauterine dissemination. This study used the STROBE statement for observational studies [6].

This study’s primary outcome was to assess the difference in the prevalence of myometrial infiltration in MSI and microsatellite stability (MSS) ECs confined to the uterus.

We also conducted a sub-analysis for the type of heterodimer leading the MSI.

### 2.3. Setting

Between August 2022 and October 2024, all patients treated for EC at the gynecologic oncology unit of the University of Campania “Luigi Vanvitelli” Naples, Italy, and the unit of obstetrics and gynecology, “Santa Maria della Misericordia” University Hospital, Udine, Italy, who met the inclusion criteria were enrolled in this retrospective observational study. All included patients had low-grade EC (Grading 1 or 2), endometrioid histotype without p53 mutations. All patients included had a total hysterectomy with bilateral salpingo-oophorectomy with bilateral pelvic node mapping (by sentinel lymph node SLN or systematic lymphadenectomy) and underwent a preoperative MRI with no evidence of extra-uterus dissemination in the 30 days previous to surgery, revised by two independent blinded expert radiologists. Patients were stratified based on the molecular profile in MSI and MSS. A sub-stratification was performed based on the loss of function of the heterodimers involved in MSI: MLH1/PMS2 heterodimer, MSH2/MSH6, or both. No follow-up was needed for the desired data.

### 2.4. Participants

The inclusion criteria to be enrolled were as follows: (1) diagnosis of endometrial endometrioid cancer G1 or G2; (2) complete anatomopathological information obtainable; (3) complete molecular profile for MLH1, PMS2, MSH2, MSH6; (4) have undergone surgical treatment with hysterectomy and at least bilateral SLN (5) for clinical stage I endometrial cancer; (6) underwent MRI within 30 days before surgery with no suspected extrauterine infiltration, confirmed by two blinded different expert radiologists. The exclusion criteria were patients with partial molecular profiling, patients with imaging suspected of advanced disease (lymph node positivity, peritoneal or parenchymal metastasis), patients with mutation of p53, patients with positive lymph nodes, and patients with an additional synchronous oncological diagnosis or within the previous 3 years.

Data were compared between patients with microsatellite stability (MSS) and microsatellite instability (MSI).

### 2.5. Variables

The variables examined were body mass index (BMI) as a continuous variable in kg/m^2^; age expressed as a continuous variable in years; grading assessed from 1 to 2, based on cytoarchitectonic alteration, judged separately by 2 pathologists, and considered as an ordinal variable; lymphovascular space invasion (LVSI), based on the absence or presence of <3 tumor cells per each LVS of more than 3 and stratified, respectively, into “negative”, “focal”, and “diffuse” [7] and considered as an ordinal variable. Myometrial infiltration was divided into “no infiltration” and “infiltration” if at least 1mm of myometrium was infiltrated by cancer, and considered a dichotomous variable. Moreover, infiltration was divided into “no infiltration”, “<50%”, and “≥50%” based on the depth of myometrial infiltration. The outcome variable of stratification was the stability of the microsatellite, defined as stable (MSS) or unstable (MSI) if at least one gene between MLH1, PMS2, MSH2, and MSH6 had a loss of expression. MSI was considered a dichotomous variable. In addition, loss of function was stratified based on the heterodimer involved in loss of function: intact expression, MLH1/PMS2 heterodimer, MSH2/MSH6, or both.

### 2.6. Laboratory

Mismatch repair (MMR) protein status was assessed by immunohistochemistry (IHC) using the following antibodies: MLH1 (clone M1, Ventana), PMS2 (clone EPR3947, Cell Marque), MSH2 (clone G219-1129, Cell Marque), and MSH6 (clone 44, Ventana). Nuclear staining of tumor cells with intensity comparable to that of internal controls (intestinal mucosa, stromal and lymphoid cells, appendix) was interpreted as retained expression, consistent with microsatellite stability (MSS, MMRp).

Given that MLH1/PMS2 and MSH2/MSH6 act as heterodimeric complexes, evaluation was performed in two steps. PMS2 and MSH6 were examined first; if either protein showed loss of staining, the corresponding partner (MLH1 or MSH2) was subsequently analyzed to distinguish between isolated and concomitant loss. Tumors were classified as microsatellite instable (MSI) or MMR deficient (MMRd) if loss of nuclear expression was observed for at least one of the four proteins. MMR proteins act as functional heterodimers (MLH1/PMS2 and MSH2/MSH6), and the loss of one partner invariably determines dysfunction of the entire complex. For this reason, we evaluated heterodimers rather than single proteins. We also collected histopathological data on myometrial infiltration, grading, and histotype, and LVSI was expressed as a semiquantitative evaluation [7]. Negative or focal involvement was considered negative. The primary outcome of interest, myometrial infiltration histopathological assessed, was expressed as a dichotomous variable, “no infiltration” and “infiltration”, defined as myometrial involvement and subclassified into “no infiltration”, “<50%”, and “≥50%”, based on the depth of myometrial infiltration. All evaluations were performed blinded by two different pathologists with more than 5 years of experience in the field. In case of discordance, a third pathologist was consulted.

### 2.7. Statistic Analysis

The sample was analyzed according to the microsatellite stability status, expressed as MSS or MSI.

Previously, all variables were analyzed using histograms and compared with parametric distributions and non-parametric distributions.

The nominal variables were expressed as absolute frequency and percentages and compared using Fisher’s exact [8] and Chi-square tests [9]. Continuous variables were expressed as median and interquartile range and compared using the Wilcoxon test [10].

The null hypothesis of our study was that there was no difference in the prevalence of myometrial infiltration between patients with MSS or MSI (H0: p1 = p2; H1: p1 − p2 ≠ 0 two-sided).

The secondary outcome was the prevalence of myometrial infiltration in the different expressions of the MLH1/PMS2 and MSH2/MSH6 heterodimer groups.

The weights of the individual values on the dichotomous dependent variable were calculated as a logistic regression [11]. The significance of the model used was assessed using the maximum likelihood method [12].

Odds ratios (ORs) and their 95% confidence intervals (CIs), obtained from univariate and multivariate logistic regression models, were graphically represented using forest plots. Each plot displayed the estimated effect size of each covariate on the presence of myometrial infiltration, with a visual emphasis on the magnitude and precision of the associations. The forest plot for the multivariate model was constructed using log-transformed ORs and diamond shapes to represent point estimates and confidence intervals. This study did not employ matching between cases and controls; all eligible patients meeting the inclusion criteria were consecutively enrolled and analyzed according to their microsatellite status. To determine whether the sample was adequate to detect a statistically significant difference in myometrial infiltration risk between the MSI and MSS groups, an “a posteriori power analysis” based on the Chi-square test for independent proportions was performed. With a significance level α = 0.05 and a true difference in the incidence of myometrial infiltration between the two groups (96% vs. 98%), the resulting statistical power (1 − β) was 65%, indicating a suboptimal probability of detecting a true association. The statistical significance level was set at 0.05. No formal sensitivity analysis was performed, as the dataset had no missing values for primary outcomes and all included patients met uniform inclusion criteria with complete molecular and histopathological data.

All statistical investigations were performed using R software and R Studio vers. 2023.12.1 + 402.

### 2.8. Risk of Bias

To reduce the risk of selection and information bias, we adopted several methodological safeguards. All included patients met strict predefined inclusion and exclusion criteria to ensure population homogeneity. Imaging evaluations were independently reviewed by two blinded expert radiologists, with discordances resolved by consensus. Likewise, histopathological assessments, including grading, LVSI, and myometrial infiltration, were performed independently by two specialized gynecological pathologists. In case of disagreement, a third senior pathologist was consulted. Molecular characterization of mismatch repair proteins (MMRs) was standardized across centers using validated immunohistochemical protocols. No matching between cases and controls was applied, but consecutive inclusion minimized selection bias. Multivariate regression studies were conducted with a combination of all variables present to minimize confounders. The individual models thus obtained were compared using adjusted R2 and Bayesian information criteria (BIC). The best model was chosen based on the lowest expressed value of BIC data analysis, conducted first by CR and then by blinding by SR, unaware of this study’s objective. No missing data were present in the outcomes of interest.

## 3. Results

Our retrospective observational cohort study, conducted at our institutions between August 2022 and October 2024, selected 217 patients and examined the MMRPs of 144 patients undergoing surgery for clinical stage I low-grade endometrial cancer, with no evidence at MRI of myometrial infiltration. The sample was stratified according to microsatellite stability: 118 MSS vs. 26 MSI. Further, patients with MSI were stratified on the basis of heterodimers involved in loss of function: 17 Loss of MLH1/PMS2, 7 Loss of MSH2/MSH6, and 2 with loss of both. The causes of inclusion and exclusion are shown in Figure 1.

### 3.1. Population Characteristics

No statistically significant differences between the two groups regarding age, BMI, ethnicity, tumor dimension, or LVSI status were found between MSI and MSS. The grading distribution was the only statistically significant difference between the two groups (G2 47% MSS vs. 81% MSI, *p* = 0.0002). All sample characteristics are shown in Table 1.

The same sample difference was also preserved in stratification according to the heterodimer involved in MSI. The group with loss of function of MLH1/PMS2 showed the highest prevalence of G2 (88%). The sub-analysis data are summarized in Table 2.

### 3.2. Outcome

This study’s main outcome was assessing the distribution of myometrial infiltration in low-grade EC MSS and MSI. The MSI and MSS patient groups showed a non-significant overlapping percentage of myometrial infiltration (98% vs. 96%; *p* = 0.5). Regarding the type of infiltration, the MSI group showed the highest probability of infiltration ≥50% of myometrial thickness (38% vs. 19%, *p* = 0.042). Data are summarized in Table 3.

### 3.3. Sub-Analysis for Heterodimer

The same analysis was conducted by stratifying the sample according to the loss of function of individual heterodimers. The prevalence of myometrial infiltration was higher in case of loss of function of MLH1/PMS2 heterodimer, compared with loss of function of MSH2/MSH6, although without reaching statistical significance (100% vs. 86%; *p* = 0.212). However, the MLH1/PMS2 group showed a statistically significant association with the risk of deep infiltration (47% vs. 14%; *p* = 0.036). These data are shown in Table 4.

### 3.4. Logit Regression

To test the correlation between the type of loss of function and myometrial infiltration, we constructed a logit regression model, with the dependent variable the presence or absence of myometrial infiltration. The MLH1/PMS2 group failed to show a statistically significant positive correlation, with an odds ratio (OR) of 1.02 (estimate 0.017, std. error 0.037; 95% CI 0.95–1.10; *p*-value 0.46). Conversely, loss of function of the MSH2/MSH6 heterodimer showed a statistically significant negative correlation with myometrial infiltration (OR 0.88, estimate −0.126, std. error 0.055; 95% CI 0.79–0.98; *p*-value 0.024). These data are reported in Table 5.

To limit the confounders, we also constructed multivariate regression models for each possible variable. Based on the lowest BIC, the best multivariate analysis model combined grading and LVSI status as confounders. In this case, the loss of MSH2/MSH6 maintained the negative association with myometrial infiltration (OR 0.88, estimate −0.124, std. error 0.052; 95% CI 0.80–0.98; *p*-value 0.020) independent of the effect of grading 2 (OR 1.05, estimate 0.048, std. error 0.024; 95% CI 1.01–1.10; *p*-value 0.042). Those data are summarized in Table 6.

The results of the multivariate logistic regression model assessing the association between MMR heterodimer loss and myometrial infiltration are visually summarized in Figure 2, using a forest plot.

## 4. Discussion

### 4.1. Interpretation of Results

The results of our study show no different incidence of myometrial infiltration in an absolute sense between patients with low grade EC (MSS and MSI). However, in patients with MSI, the risk of deep myometrial infiltration (≥50% of myometrial thickness) is higher. When the sample is stratified by type of loss of function that determines MSI status, the heterodimer MLH1/PMS2 loss is more likely to involve deep infiltration. This finding is also supported by the evidence that in the case of the loss of both heterodimers, the risk is increased compared with the loss of MSH2/MSH6 alone. Logistic regression models, however, do not show a strong correlation between the loss of function of this heterodimer and myometrial infiltration. However, in contrast, loss of function of the MSH2/MSH6 heterodimer shows a protective value against myometrial infiltration in a context of MSI. This feature is also confirmed in light of the weight exerted by the grading and involvement of LVIS. It should be emphasized that patients with clinical suspicion of uterine-confined tumors represent the population under investigation. Therefore, our study emphasizes how microsatellite instability favors tumor progression even in this category of patients, who are always considered “slow-moving” [13,14,15]. The greater ease of progression of MSI tumors could be justified at the level of the tumor microenvironment. Recently, studies have shown how microsatellite instability may correlate with the expression of immune checkpoints [16,17] and the upregulation of miRNAs that may favor tumorigenesis [18]. The mutation state should, therefore, accelerate tumor degeneration, making even low-grade types potentially more aggressive than their MSS counterparts.

### 4.2. Clinical Implications

The information about MS status, as demonstrated in our study, even in the presence of “clinically reassuring” diseases such as confined uterine disease and low-grade endometrioid histotypes, adds valuable information to the therapeutic planning of these patients. The greater likelihood of these patients presenting with myometrial infiltration can influence the therapeutic and staging approach. Finally, recent insights into the molecular profile of endometrial carcinomas have opened new horizons to better characterize the prognostic impact of the different presentations of this tumor [3,19,19,20]. The findings of our study carry relevant clinical implications, especially in the context of preoperative counseling and treatment planning for patients with low-grade endometrioid EC. Although these tumors are generally considered indolent, our data suggest that MSI, particularly with MLH1/PMS2 loss, is associated with a significantly higher risk of deep myometrial infiltration, despite reassuring imaging. MRI remains the gold standard for preoperative staging, although not perfect. Our results suggest that MSI status might serve as a molecular red flag in otherwise low-risk patients, providing complementary—rather than substitutive—information. In addition, MRI is an expensive and difficult-to-organize method, especially in developing countries, so more information about the risk of myometrial infiltration can help select cases for further investigation and optimize the allocation of health care resources. This information could support a more cautious selection for conservative management and inform the need for more extensive staging procedures. MSI status might be considered as additional information for personalized decision making, but routine preoperative MSI screening cannot be recommended at present and requires prospective validation. This is all the more useful with the increasing number of diagnoses in patients who have not completed their reproductive cycle [21]. Fertility-sparing treatments are reserved for those with low-grade carcinomas and in the absence of myometrial infiltration [22,23,24,25]. However, the absence of myometrial infiltration remains presumptive; therefore, adding the mutational status as information can help direct the right treatment choices. Our study fits into this mosaic as one more piece to personalize the surveillance of patients with EC.

### 4.3. Comparison with the Literature

Our findings align with the existing literature emphasizing the prognostic significance of microsatellite instability (MSI) in endometrial carcinoma. Studies by Stelloo et al. [26] and Leon-Castillo et al. [27] have demonstrated that MSI-positive tumors exhibit distinct molecular profiles and often present more aggressive behavior compared with microsatellite-stable (MSS) counterparts. Similarly, our study corroborates these observations by showing a higher prevalence of myometrial infiltration in the MSI group, even in clinically low-risk patients. Furthermore, MSI tumors are suggested to have a unique tumor microenvironment characterized by upregulated immune checkpoint expressions, which may partially explain their higher propensity for progression. In contrast, studies such as that by Liu et al. [28] have highlighted the potential therapeutic vulnerabilities of MSI-positive tumors, particularly their responsiveness to immune checkpoint inhibitors [29,30,31,32]. In addition, this research group recently demonstrated that tumor progression at the microscopic level triggers an immune response. Tumor progression, either as an advancement in grading or myometrial infiltration, triggers a response from the immune system [33,34]. Finally, recently, our research group showed that patients with MSI have a higher risk of presenting with occult non-clinically evident lymph node metastasis [35]. Similar to the findings in this second patient setting, loss of the MLH1/PMS2 heterodimer is more likely to be associated with tumor progression than the MSH2/MSH6 heterodimer. In this context, the finding from our study that myometrial infiltration is more frequent in MSI endometrial cancers is not unexpected. A possible explanation lies in the well-documented genomic plasticity of MSI tumors, which may enhance their ability to adapt and progress [36]. The presence of microsatellite instability may promote the mechanisms of tumor escaping from the immune system [16,17,37,38]. Our data support this therapeutic relevance by reinforcing the need for routine assessment of MSI status, even in ostensibly indolent cases. These comparisons underscore the growing interest that MSI is not only a biomarker of progression but also a pivotal determinant in therapeutic and prognostic stratification.

### 4.4. Strenght and Limitations

The limitation of our study is its retrospective nature, which increases the possibility of inclusion bias. On the contrary, however, the strict inclusion criteria and having limited the enrolment to only patients with low-grade carcinoma and in the absence of clinic suspicion of extra-uterine disease helps to give a fair correlation with the effect exerted by microsatellite instability in the progression of this pathology. Another limitation of our study is the absence of an a priori sample size calculation. Although we performed a post hoc power analysis to assess the ability of our sample to detect significant differences between groups, this approach is methodologically weaker. It does not substitute a formal pre-study justification of sample size. Nevertheless, this study is methodologically robust, with well-defined and rigorous inclusion and exclusion criteria. Furthermore, the measures adopted to reduce bias enhanced the reliability of the findings, providing strong statistical support. Further prospective studies and validation of patients’ DNA to confirm the molecular status will be necessary to clarify doubts. Moreover, no formal sensitivity analysis was performed, as the dataset had no missing values for primary outcomes and all included patients met uniform inclusion criteria with complete molecular and histopathological data. A further limitation of our study is the relatively small number of patients with MSI (n = 26), which reduces the generalizability of our findings and underlines the need for validation in larger prospective cohorts. Given the sample size and the homogeneity of the population, we deemed the risk of substantial bias from unmeasured confounding to be limited. However, we acknowledge that further sensitivity analyses in larger cohorts may strengthen the robustness of our findings. Finally, a further limitation is the absence of prognostic data, given the proximity to the enrolment period. Future follow-ups on the sample will also make deducing any differences in prognostic terms possible.

### 4.5. Future Prospectives

Future studies should aim to validate these findings in larger prospective cohorts, ideally incorporating long-term oncologic and reproductive outcomes. In particular, assessing the prognostic significance of MSI and specific MMR heterodimer loss in relation to recurrence and survival would help establish their role as independent biomarkers. Moreover, integrating MSI status with imaging and molecular classifiers, such as ProMisE or TCGA-based models, may enhance preoperative risk stratification. Finally, prospective trials evaluating the safety and outcomes of fertility-sparing treatment in MSI-positive patients, even in the clinical absence of myometrial infiltration, could help define more precise selection criteria for conservative management in young women.

## 5. Conclusions

In low-grade endometrial carcinomas, clinically confined to the uterine body, MSI, based on the loss of function of MLH1/PMS2, increases the likelihood of myometrial infiltration. Our results suggest that MSI, particularly with MLH1/PMS2 loss, may be associated with a higher likelihood of deep myometrial infiltration. These findings should be considered hypothesis-generating and require confirmation in larger prospective studies. Future studies will be necessary to assess the prognostic weight of these findings.

## Figures and Tables

**Figure 1 jpm-15-00417-f001:**
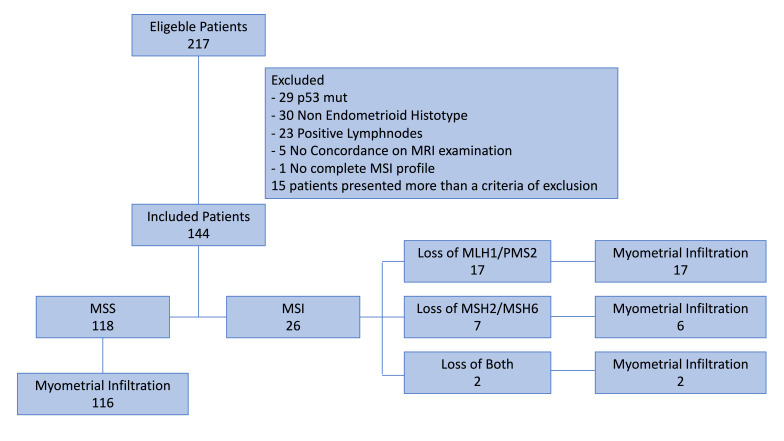
Enrollment flow chart.

**Figure 2 jpm-15-00417-f002:**
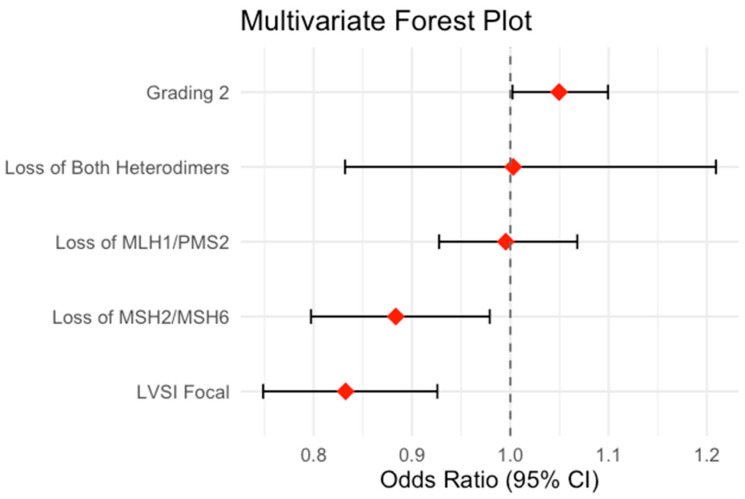
Forrest plot.

**Table 1 jpm-15-00417-t001:** Patients’ characteristics.

Characteristic	MSI, N = 26 ^1^	MSS, N = 118 ^1^	*p*-Value ^2^
Age	66 (60, 76)	62 (56, 71)	0.066
BMI	29 (27, 33)	29 (25, 33)	0.979
Missing	0	1	
Ethnicity			>0.999
Caucasian	26 (100%)	116 (98%)	
Hispanic	0 (0%)	1 (0.8%)	
Indian	0 (0%)	1 (0.8%)	
Grading			0.002
1	5 (19%)	62 (53%)	
2	21 (81%)	56 (47%)	
LVSI			>0.999
Negative	21 (81%)	96 (82%)	
Diffuse	3 (12%)	13 (11%)	
Focal	2 (7.7%)	8 (6.8%)	
Missing	0	1	
Dimension	20 (14, 25)	20 (10, 30)	0.879
Missing	13	77	
Lymph Node Retrieved	2.00 (2.00, 4.75)	2.00 (2.00, 4.00)	0.599

^1^ Median (IQR); N (%). ^2^ Wilcoxon rank sum test; Fisher’s exact test.

**Table 2 jpm-15-00417-t002:** Patients’ characteristics based on heterodimer.

Characteristic	Intact Nuclear Expression, N = 118 ^1^	Loss of Both Heterodimers, N = 2 ^1^	Loss of MLH1/PMS2, N = 17 ^1^	Loss of MSH2/MSH6, N = 7 ^1^	*p*-Value ^2^
Age	62 (56, 71)	70 (65, 74)	66 (59, 76)	68 (62, 71)	
BMI	29 (25, 33)	32 (30, 34)	28 (26, 31)	33 (29, 35)	
Missing	1	0	0	0	
Ethnicity					>0.999
Caucasian	116 (98%)	2 (100%)	17 (100%)	7 (100%)	
Hispanic	1 (0.8%)	0 (0%)	0 (0%)	0 (0%)	
Indian	1 (0.8%)	0 (0%)	0 (0%)	0 (0%)	
Grading					0.004
1	62 (53%)	1 (50%)	2 (12%)	2 (29%)	
2	56 (47%)	1 (50%)	15 (88%)	5 (71%)	
LVSI					0.822
Negative	96 (82%)	2 (100%)	14 (82%)	5 (71%)	
Focal	8 (6.8%)	0 (0%)	1 (5.9%)	1 (14%)	
Diffuse	13 (11%)	0 (0%)	2 (12%)	1 (14%)	
Missing	1	0	0	0	
Dimension	20 (10, 30)	NA (NA, NA)	20 (15, 25)	30 (22, 37)	
Missing	77	2	6	5	
Lymph Nodes Retrieved	2.00 (2.00, 4.00)	2.00 (2.00, 2.00)	3.00 (2.00, 5.00)	2.00 (2.00, 2.50)	

^1^ Median (IQR); N (%). ^2^ Fisher’s exact test. NA: Not Avaivable.

**Table 3 jpm-15-00417-t003:** Outcomes.

Characteristic	MSI, N = 26 ^1^	MSS, N = 118 ^1^	*p*-Value ^2^
Myometrial Infiltration			0.5
No Infiltration	1, (3.8%)	2, (1.7%)	
Infiltration	25, (96%)	116, (98%)	
Type of Infiltration			0.042
No Infiltration	1, (3.8%)	2, (1.7%)	
<50%	15, (58%)	94, (80%)	
≥50%	10, (38%)	22, (19%)	

^1^ N, (%). ^2^ Fisher’s exact test.

**Table 4 jpm-15-00417-t004:** Outcomes based on heterodimers.

Characteristic.	Intact Nuclear Expression, N = 118 ^1^	Loss of Both Heterodimers, N = 2 ^1^	Loss of MLH1/PMS2, N = 17 ^1^	Loss of MSH2/MSH6, N = 7 ^1^	*p*-Value ^2^
Myometrial Infiltration					0.212
No Infiltration	2 (1.7%)	0 (0%)	0 (0%)	1 (14%)	
Infiltration	116 (98%)	2 (100%)	17 (100%)	6 (86%)	
Type of Infiltration					0.036
No Infiltration	2 (1.7%)	0 (0%)	0 (0%)	1 (14%)	
<50%	94 (80%)	1 (50%)	9 (53%)	5 (71%)	
≥50%	22 (19%)	1 (50%)	8 (47%)	1 (14%)	

^1^ N (%). ^2^ Fisher’s exact test.

**Table 5 jpm-15-00417-t005:** Logit regression. Logistic regression heterodimer on myometrial infiltration.

Variable	Estimate	Std. Error	z Value	*p*-Value	Odds Ratio	OR 95% CI
Loss of Both Heterodimers	0.017	0.101	0.167	0.867	1.0171453	0.834–1.24
Loss of MLH1/PMS2	0.017	0.037	0.460	0.646	1.0171453	0.946–1.093
Loss of MSH2/MSH6	−0.126	0.055	−2.279	0.024	0.8816148	0.791–0.982

**Table 6 jpm-15-00417-t006:** Multivariate logit regression. Logistic multivariate regression.

Variable	Estimate	Std. Error	t Value	*p*-Value	Odds Ratio	OR 95% CI
Loss of Both Heterodimers	0.003	0.095	0.031	0.975	1.0030045	0.832–1.209
Loss of MLH1/PMS2	−0.005	0.036	−0.129	0.898	0.9950125	0.928–1.068
Loss of MSH2/MSH6	−0.124	0.052	−2.364	0.020	0.8833798	0.797–0.979
Grading 2	0.048	0.024	2.053	0.042	1.0491707	1.002–1.1
LVSI Focal	−0.184	0.044	−4.173	0.001	0.8319358	0.763–0.907
LVSI Diffuse	−0.001	0.036	−0.023	0.981	0.9990005	0.931–1.073

## Data Availability

The original data presented in the study are available in 10.5281/zenodo.15520679 or on request from the corresponding author.

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
