# Peer review of "Microsatellite Instability and Myometrial Infiltration in Low-Grade Endometrial Cancer: A Focus on MMR Heterodimer Dysfunction by a Retrospective Multicentric Italian Study"

_jpm, 2025, doi:10.3390/jpm15090417_

Round 1

Reviewer 1 Report

Comments and Suggestions for Authors

It’s a very well-concerned article that highlights the possible role of MSI in the evaluation of myometrial infiltration for endometrial cancer(EC) limited to the uterus(low risk).The study follows all the steps of a serious research,with clear and complete informations about the subject.The Results are clearly expressed,through elaborate and clarifying tables regarding all aspects of the study group.However,the Clinical Implication subchapter has exaggerated statements,since serious future researches are needed to confirm the authors’results.MRI represents the best current tool or preoperative assessment of myometrial infiltration and although it’s  not perfect,it cannot be replaced for now by MSI status.The presence of MSI can be considered only a red flag,but it cannot modify either the therapeutic management or the follow-up of cases with low-risk EC.Also at the end of tha Comparison with Literature sub chapter,the authors state the existence of a growing consensus for the introduction of MSI status as a biomarker of progression and a pivotal determinant in therapeutic and prognostic stratification of low-risk EC,which is an exaggerated statement at this moment,a wish but not a reality.Fortnately,the Conclusions are short and to the point and the authors return to reality.The English language is fluent and the References are well selected and correctly cited.

Author Response

We sincerely thank the Reviewer for the careful and constructive comments, which we have found very helpful in refining our manuscript. We fully agree that the clinical implications of our findings must be interpreted with caution, and we have revised the relevant sections accordingly. Below we provide a point-by-point reply:

  1. On the Clinical Implication section
    We acknowledge the Reviewer’s concern that some statements were overstated. Our study was retrospective and exploratory, and therefore cannot provide definitive evidence to change clinical management. In the revised version, we have rephrased the section to underline that MSI status may represent a potential molecular red flag in otherwise low-risk cases, but it cannot replace MRI in preoperative staging nor independently dictate therapeutic or follow-up strategies at present. We now explicitly state that prospective validation is needed before any clinical translation.

  2. On the role of MRI vs MSI
    We agree that MRI remains the current gold standard for preoperative evaluation of myometrial infiltration, despite its limitations. In the revision, we clarify that MSI status is not an alternative to MRI but could provide complementary information in selected cases, particularly when fertility-sparing management is under discussion. This nuance has been made explicit.

  3. On the statement about a “growing consensus”
    We thank the Reviewer for pointing out that this wording may suggest a stronger level of agreement than currently exists. We have modified the phrasing to indicate that there is increasing interest in the role of MSI, while consensus is still lacking. We emphasize that MSI testing is being studied as a prognostic and therapeutic biomarker, but its introduction into routine practice for low-risk EC remains a perspective, not a reality.

  4. On the Conclusions
    We are pleased the Reviewer appreciated their conciseness and balance. We have kept them short and factual, highlighting the exploratory nature of our results and the need for larger prospective studies.

  5. Language and References
    We thank the Reviewer for acknowledging the fluency of the language and the adequacy of the references.

Reviewer 2 Report

Comments and Suggestions for Authors

Congratulations to the authors for choosing this interesting topic. The article is well structured and organised.

I have some recommendations to improve the article:

The group of MSI is much smaller (26 patients) – you should discuss this as limitation

Please briefly describe in the introduction why you chose MMR proteins and not p53 or POLE.

Why did you select the heterodimers MLH1/PMS2 and MSH2/MSH6 from these proteins and not all proteins alone?

In table 1, you have significant difference in grading. In MSI, 81% of patients have grade 2, which may falsify your results. Please mention this in the results and discuss it.

Author Response

We sincerely thank the Reviewer for the constructive suggestions, which have been very helpful to improve the quality of our manuscript. Below we address each comment:

  1. Small MSI group (26 patients)
    We agree with the Reviewer that the MSI group is relatively small. We have now explicitly mentioned this in the Limitations section, clarifying that the reduced sample size limits the generalizability of our findings and highlights the need for validation in larger, prospective cohorts.

  2. Choice of MMR proteins instead of p53 or POLE
    Our study specifically focused on mismatch repair deficiency (MMRd) as a molecular driver of microsatellite instability (MSI). Including p53 or POLE mutations would have introduced a confounder, since they represent other molecular categories according to TCGA/ProMisE classification and follow distinct biological and clinical pathways. For this reason, we deliberately limited the analysis to MMR proteins (MLH1, PMS2, MSH2, MSH6). We have now clarified this rationale in the Introduction.

  3. Selection of heterodimers MLH1/PMS2 and MSH2/MSH6 instead of single proteins
    We thank the Reviewer for this important point. From a biological and clinical perspective, the functionality of MMR proteins relies on heterodimer formation. Loss of one protein within the pair invariably determines dysfunction of the entire heterodimer, making the evaluation of individual proteins not clinically meaningful. For this reason, we analyzed MLH1/PMS2 and MSH2/MSH6 as functional units, which better reflects their biological role. This has now been clarified in the Methods.

  4. Difference in grading distribution (Table 1)
    We acknowledge that the MSI group had a significantly higher proportion of grade 2 tumors (81%). This difference is clinically expected, as MSI tumors are frequently associated with higher grade. Importantly, our multivariate analysis already accounted for grading as a potential confounder and demonstrated that the association between MMR heterodimer loss and myometrial infiltration remained significant, particularly for MSH2/MSH6. Therefore, while grading plays a role in infiltration risk, it does not invalidate our results, which suggest an independent association with the missing heterodimer.